# On-Site Pilot Testing of Hospital Wastewater Ozonation to Reduce Pharmaceutical Residues and Antibiotic-Resistant Bacteria

**DOI:** 10.3390/antibiotics10060684

**Published:** 2021-06-08

**Authors:** Sofia Svebrant, Robert Spörndly, Richard H. Lindberg, Therese Olsen Sköldstam, Jim Larsson, Patrik Öhagen, Hanna Söderström Lindström, Josef D. Järhult

**Affiliations:** 1Uppsala University Hospital, Uppsala County Council, 75185 Uppsala, Sweden; sofia.svebrant@akademiska.se (S.S.); therese.olsen@akademiska.se (T.O.S.); jim.larsson@regionuppsala.se (J.L.); 2Zoonosis Science Center, Department of Medical Sciences, Uppsala University, 75185 Uppsala, Sweden; robert.sporndly@medsci.uu.se; 3Department of Chemistry, Umeå University, 90187 Umeå, Sweden; richard.lindberg@umu.se; 4Uppsala Clinical Research Center, Uppsala University, 75185 Uppsala, Sweden; patrik.ohagen@ucr.uu.se; 5Section of Sustainable Health, Department of Public Health and Clinical Medicine, Umeå University, 90187 Umeå, Sweden; hanna.soderstrom@umu.se

**Keywords:** antibiotic resistance, API, sewage treatment, drug residues, environment, ozone treatment, pharmaceuticals

## Abstract

Hospital sewage constitutes an important point source for antibiotics and antibiotic-resistant bacteria due to the high antibiotic use. Antibiotic resistance can develop and cause problems in sewage systems within hospitals and municipal wastewater treatment plants, thus, interventions to treat hospital sewage on-site are important. Ozonation has proven effective in treating relatively clean wastewater, but the effect on untreated wastewater is unclear. Therefore, we piloted implementation of ozonation to treat wastewater in a tertiary hospital in Uppsala, Sweden. We measured active pharmaceutical ingredients (APIs) using liquid chromatography-mass spectrometry and antibiotic-resistant Enterobacteriaceae using selective culturing pre- and post-ozonation. Comparing low (1 m^3^/h) and high (2 m^3^/h) flow, we obtained a ‘dose-dependent’ effect of API reduction (significant reduction of 12/29 APIs using low and 2/29 APIs using high flow, and a mean reduction of antibiotics of 41% using low vs. 6% using high flow, 25% vs. 6% for all APIs). There was no significant difference in the amount of antibiotic-resistant Enterobacteiaceae pre- and post-ozonation. Our results demonstrate that ozonation of untreated wastewater can reduce API content. However, due to the moderate API decrease and numerous practical challenges in the on-site setting, this specific ozonation system is not suitable to implement at full scale in our hospital.

## 1. Introduction

Pharmaceutical residues (or active pharmaceutical ingredients, APIs) can reach the aquatic environment through discharge from wastewater treatment plants (WWTPs) [1]. Patients using pharmaceuticals excrete APIs via urine and, to some degree, feces, and thus wastewater contains APIs originating from patients [2]. Today, most WWTPs are not designed to remove APIs, and most of these substances pass the plant unaffected [3]. In the recipient, APIs became a risk to both aquatic and terrestrial ecosystems and, in the long run, also a health risk to humans [4]. There is also a risk of antibiotic resistance spread in the environment, even at low concentrations, or antibiotic resistance development in sewage systems and WWTPs [5,6].

Multiple studies have demonstrated that ozonation as an extra treatment step in WWTPs can reduce APIs in the effluent [7,8,9,10,11]. Ozone reacts non-selectively with pharmaceuticals in a two-step process: Direct oxidation of molecules and indirect by produced hydroxyl (OH^−^) radicals. The hydroxyl radicals may react with APIs that are not directly affected by ozone [7]. The effectiveness of ozone treatment depends on various factors, but the matrix of the wastewater is important [12]. For example, dissolved organic carbon (DOC), suspended solids, chemical oxygen demand, pH, nitrate bromide, and bromate are important [13]. Bromide is a critical compound in wastewater ozonation as it can be oxidized to carcinogenic bromate [13]. When using oxidative treatment for pharmaceutical removal, there is also a risk for transformation products from the parent substance that can be more toxic than the parent substance [14].

Due to the high antibiotic use, hospital wastewater contains high levels of APIs, including antibiotics (among which are last-line drugs), antibiotic-resistant bacteria, and wild-type bacteria that can serve as recipients for horizontal transfer of resistance. Today, a vast majority of hospitals do not treat their sewage water, and the sewage is discharged to municipal WWTPs as is [15]. Resistance can be problematic inside hospitals and sewage systems as effluent wastewater contains resistant bacteria [16] and their genes can be transferred to pathogenic bacteria and spread inside the hospital through sinks, toilets, and/or air [15]. A study comparing different techniques to treat hospital wastewater concluded that an MBR (membrane bioreactor) combined with an ozonation step is the most cost-efficient method to remove APIs and antibiotic-resistant bacteria [10]. At Copenhagen university hospital in Denmark, a full-scale WWTP using MBR and ozonation is installed as the sewage is then discharged directly to a recipient [17].

Most studies assessing ozonation of wastewater for reduction of pharmaceuticals focus on pre-treated wastewater (often in late stages of WWTP treatment) that is low in DOC and relatively clean and similar to water discharged to the recipient [7]. However, hospitals can serve as a point source for the spread of antibiotic resistance as: (i) A high antibiotic use gives rise to a high concentration of antibiotics in the sewage [10], (ii) several especially important, last-line antibiotics are only used in hospitals, and (iii) high antibiotic use leads to resistance development of enteric bacteria of patients, and these antibiotic-resistant bacteria end up in the sewage. These factors can lead to resistance development already within the wastewater system as such, in the hospital, in sewage pipes, and inside the WWTP [18]. Therefore, there is a need for measures to reduce levels of antibiotic residues and antibiotic-resistant bacteria at or close to the hospital before discharging the wastewater to the sewage system and the municipal WWTP. Thus, the aim of this study was to pilot on-site implementation of an ozonation step for sewage treatment at a tertiary hospital in Sweden. We demonstrate that ozonation resulted in a significant reduction of levels of several APIs, illustrating that it is possible to reduce the amount of drug residues in hospital sewage by ozonation of untreated wastewater. However, there were severe practical challenges in the implementation, a relatively low reduction in API levels, and no obvious reduction in antibiotic-resistant bacteria, and therefore, we do not find this particular ozonation system useful in our hospital setting, although the technique per se may be useful.

## 2. Results

### 2.1. Drug Residues

Of the 92 APIs that were analyzed by liquid chromatography-mass spectrometry (a full list of analyzed APIs and their limits of quantification (LOQs) are given in Appendix A), 34 were detected in at least one sample. Further analysis of data was performed on the 29 APIs where a maximum of 40% of the measurements was <LOQ. These results are presented in Table 1. In general, there was a high variation in API levels in between sampling occasions, likely due to the fact that grab samples were used. Using low flow (1 m^3^/h, allowing a longer contact time between wastewater and ozone = more ozonation effect), 12 out of the 29 APIs had a significantly lower level post-ozonation (sampling point 3, see Figure 1) as compared to pre-ozonation (sampling point 2). High flow (2 m^3^/h) resulted in two APIs with significantly lower levels post-ozonation as compared to pre-ozonation.

To obtain an estimate of reduction in API levels, the difference post-ozonation—pre-ozonation was calculated for each API and sampling occasion. The median of these differences for all APIs is presented in Table 1. The average API reduction (calculated as mean of all API differences) was 25% (low flow) and 6% (high flow) for all APIs, and 41% (low flow) and 6% (high flow) for the five antibiotics included among APIs.

### 2.2. Antibiotic-Resistant Bacteria

One set of triplicate samples obtained at low flow conditions was analyzed on agar plates selective for ESBL-producing Enterobacteriaceae. The results demonstrated no significant reduction in cfu/mL, pre-ozonation a mean of 14,500 cfu/mL (SD 4910 cfu/mL) was detected and post-ozonation a mean of 12,300 cfu/mL (SD 1160 cfu/mL).

### 2.3. Energy Consumption of Ozonation System

The total energy consumption of the ozonation step measured for the 63 days when the system was functional was 342 kWh. During this time, the ozonation step was active for approximately 2.5 h per day. Thus, we estimate the energy consumption of the ozonation step to be 19 MWh/year of full activity of the system.

## 3. Discussion

Our on-site pilot study results demonstrate that ozonation of untreated hospital wastewater can significantly reduce levels of APIs. The results indicate that the ozonation effect was significantly increased in the low flow setting as compared to high flow, indicating a ‘dose-dependent’ effect in the sense that running the system at lower flow (1 m^3^/h vs 2 m^3^/h), allowing more reaction time for the ozone per unit of water, gave rise to a significant reduction of more APIs (12 APIs vs. 2 APIs). Due to the large variation in API levels, it was hard to evaluate the reduction in level for each individual API that spanned from 60% reduction to 2% increase (low flow) and 67% reduction to 13% increase (high flow). We therefore calculated the mean reduction for all APIs for each flow setting to roughly estimate the total API reducing capability. The same calculations were made for antibiotics only. The results further support a ‘dose-dependent’ effect as a low flow led to a higher average API reduction than high flow (41% vs. 6% for antibiotics, 25% vs. 5% for all APIs). The average API level reductions even for the lower flow are relatively low in comparison to previous studies on pre-treated wastewater that have demonstrated reductions up to 80–90% [9,13].

Likely, the high API level variation in between sampling occasions led to an underestimation of the number of significant reductions of API levels. This problem may have been reduced if continuous samplings could have been performed (e.g., for 24 h) to reduce short-term variations. Unfortunately, this was not possible in our study setting due to practical challenges with the pilot system (discussed in more detail below). In addition, API measurements in matrix-rich samples such as untreated sewage waters usually come with a higher degree of uncertainty. The statistical analysis of API levels involved numerous statistical tests (one for each API and flow), which carries a risk of type I errors. However, relatively many significant differences were found (14 significant differences out of 58 tests), and all significant differences were related to API reduction (significant differences due to a type I error would have been evenly distributed between reductions and increased levels), strongly indicating a biological significance of the statistical findings.

The culturing for antibiotic-resistant bacteria did not demonstrate any significant difference pre- and post-ozonation. This was somewhat surprising, as ozonation is known to efficiently kill bacteria, and lab-scale pilot experiments of a similar system have shown a rapid and marked decrease in live coliform bacteria (data not shown). A limitation of our study is that, for practical reasons, only one triplicate sampling was performed. However, the sampling was performed when the system was operating at the more efficient low flow setting (1 m^3^/h), and therefore, we argue that the expected clear reduction in load of antibiotic-resistant bacteria should have been detectable even during our limited sampling. Furthermore, two more identical triplicate samplings done prior to the one described in this paper at a similar flow did not show any difference in antibiotic-resistant bacterial load (data not shown), which supports the absence of a clear reduction effect of the ozonation step even if we decided not to include the results in the formal analysis as there were some doubts as to whether the flow in the ozonation system was perfectly calibrated.

The study hospital, UUH, is located in an area that is important for the drinking water supply to the city (Uppsala esker), and thus, it would not be an option to build a full-scale WWTP near the hospital. Therefore, we investigate other techniques to treat the hospital wastewater in order to reduce APIs in general and antibiotics in particular, and antibiotic-resistant bacteria—as exemplified by the present study. An advantage of implementing an additional sewage treatment step at the hospital is that after this step, the wastewater is discharged to the municipal WWTP, meaning that potentially harmful by-products are not released directly to the recipient. Implementing sewage treatment in a hospital area is a practical challenge. The system must be possible to accommodate without a disturbing smell and should be relatively easy to maintain by the hospital technical staff. In this regard, the tested system had several drawbacks. Although a 0.8 mm drum filter was used before the water entered the ozonation step, enough particles passed the filter to quickly clog parts of the system, necessitating frequent backwashes and cleanings. Pumps and piping had to be readjusted to accommodate for particles in the water and low flow rates. Other practical challenges faced were leakage of ozone from the collecting tank and unforeseen changes of pressure in the system. Although we had the pilot system in place for one year, the practical challenges only allowed us to run the system for approximately two months in the planned way, and then only for periods of time while being actively supervised and maintained by a technician. Regarding the ozonation step, a drawback of this study is that an exact ozone dose was not possible to obtain, which makes a more detailed evaluation of the ozonation process difficult. The energy consumption of the ozonation step (estimated to 19 MWh/year of full activity) was reasonably low, and not a major factor in the cost aspect.

Although the ozonation system was able to reduce the levels of several APIs, especially in the low flow setting, we do not find this particular system implementable at the study hospital. This is due to the numerous practical challenges faced, and also due to the relatively low reduction of API/antibiotic levels and absence of reduction of antibiotic-resistant bacteria. However, the proof-of-concept result from our study that ozonation of untreated hospital wastewater can significantly reduce API/antibiotic levels warrants further studies on the subject using systems more tailored to the on-site hospital setting.

## 4. Materials and Methods

### 4.1. Study Site

Uppsala University Hospital (UUH) is a tertiary hospital with 1000 beds of which around 600 were in use during the study period. In the hospital area, there are three cesspits of which the one used in this study accounts for the majority of sewage flow (average flow approximately 25 m^3^/h), and to which most of the hospital wards are connected. These wards include several ones with high antibiotic use, such as infectious diseases, oncology, and intensive care.

### 4.2. Experimental Setup

The pilot-scale ozonation system (RENA vivo C-series, OzoneTech, Stockholm, Sweden) was installed in a container and placed close to the chosen sampling cesspit outside the hospital. One to two m^3^ of wastewater per hour were collected, corresponding to approximately 4–8% of the total average flow in this cesspit. Figure 1 shows a schematic of the entire pilot-scale system. A submerged pump (Xylem DXGM 25-11) in the cesspit fed water to a container that contained the entire pilot system. First, the wastewater was filtered through a 0.8 mm drum filter (Roto-Sieve RS11) and then pumped (using a Grundfos CME3-2 A-R-A-E-AQQE pump) into a 1 m^3^ collecting tank and further into the 300 L oxidation reaction tank. Ozone was produced via corona discharge ozone generators and feed gas was provided by oxygen generators. A fraction of the incoming water to the reactor tank was pumped through a venturi injector to dissolve ozone in the water. Pressure sensors were installed at the top of the reaction tank, and at both sides of the venturi injector. A safety pressure valve was also installed at the top of the tank. The pressure pre ozone injection was 0.8 bar and 0.1 bar post-injection. This was pre-determined by the manufacturer. The pressure in the reaction tank during operation was 0–0.1 bar. An oxidation reduction potential (ORP) sensor and a flow meter were installed at the outlet of the oxidation reactor tank. The water flow was controlled manually by the first pump in the system and the hydraulic retention time (HRT) in the reactor tank was calculated with readings from the flowmeter and the known volume of the reactor tank (300 L) as HRT = 300 L/Flow (L/h). Two different flow settings were used: ‘Low flow’ = 1 m^3^/h resulting in a HRT = 300/1000 = 0.3 h = 18 min, and ‘high flow’ = 2 m^3^/h resulting in a HRT = 300/2000 = 0.15 h = 9 min.

Residual ozone was degassed from the system to an active carbon ozone destructor. As the treated wastewater left the system, it was used to flush out filter waste from the drum filter back to the hospital sewage cesspit. The electricity consumption of the ozone generator was measured using an electricity meter (GARO, type GNM3D).

### 4.3. Sampling

For this study, sampling was performed at sampling points 2 and 3 as 45 mL grab samples that were immediately frozen at −20. Sampling was performed during two different flow regimes, low flow and high flow, with corresponding HRTs, as calculated above. Sampling points 1 and 4 were not used in this study. The different flow regimes were set up with the pump from the collecting tank to the oxidation reactor tank. When switching from one flow regime to another, the system was left to stabilize for a minimum of one hour.

### 4.4. Analysis of Pharmaceuticals

Ninety-two APIs were analyzed with an online solid phase extraction/liquid chromatography tandem mass spectrometry (online SPE/LC-MS/MS) system and a method previously described in detail [19]. Specific details on the online SPE/LC system and the MS/MS transition ions used are given in [20] and [21], respectively. The transitions of the following APIs have not been described before: Propranolol, LOQ 30 ng/L, 260.01 → 183.2 (quantification ion (Qi)), collision energy (CE) 18, tube lens (TL) 90 and 260.01 → 155.2 (qualification ion (qi)), CE 26, TL 90; and cetirizine, LOQ 15 ng/L, 389.1 → 201.1 Qi, CE 18, TL 104 and 289.1 → 166.1 qi, CE 37, TL 104.

### 4.5. Sampling for and Analysis of Antibiotic-Resistant Bacteria

On one occasion, triplicate samples (20 min in between replicates) were obtained at sampling points 2 and 3, similarly as for APIs described above, during 1 m^3^/h flow in the system (low flow). These samples were transported to the lab on ice, diluted ted-fold, and plated on CHROMAGAR C3G plates. The C3G plates are selective for ESBL-producing Enterobacteriaceae. Plates were incubated at 37 °C overnight and then colonies were counted manually. Using 0.1 mL inoculum and 10-fold dilution, CFU/mL can be calculated as observed colonies × 100.

### 4.6. Statistics

Due to non-normal distribution of API levels, average levels were expressed as medians, and the non-parametric signed rank test was used to test for significant differences. API levels < LOQ were imputed as 1 ng/L for statistical analysis. Differences with a <5% probability of a type I error (i.e., *p* < 0.05) were considered significant. No adjustments for multiplicity were performed, hence, *p*-values have to be interpreted with some caution. The SAS program was used for statistical analysis.

## 5. Conclusions

Ozonation of untreated hospital wastewater can significantly reduce levels of antibiotics and other APIs, but in our study setting, the reduction was relatively low, and there were numerous practical challenges in implementing the system in the on-site hospital setting.

## Figures and Tables

**Figure 1 antibiotics-10-00684-f001:**
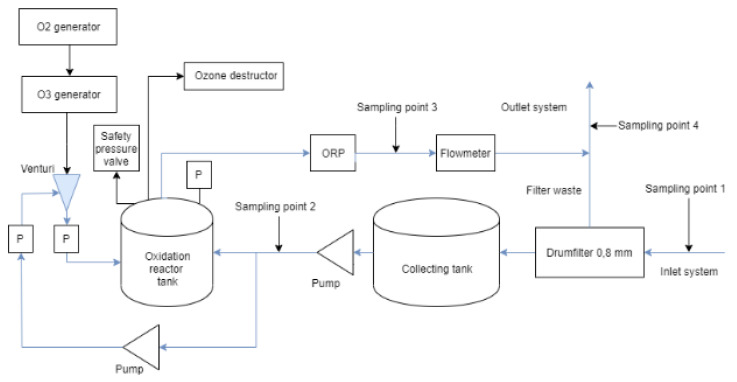
Schematic picture of the pilot-scale system inside the container. The water inlet is from the right in the picture, then it passes through a drum filter before entering the collection tank. From the collection tank, the water goes to the oxidation reactor tank. The treated water is used to flush out filter waste from the drum filter from the container. It also shows the sampling points 1–4. “P” in the figure means pressure sensor and ORP is an in-line sampling point for oxidation-reduction potential.

**Table 1 antibiotics-10-00684-t001:** Median API level differences in ng/L. Differences calculated as post-ozonation—pre-ozonation for each API and sampling occasion. Percent change compared to pre-ozonation levels given for all APIs. Statistical testing preformed using the signed rank test. LOQ = limit of quantification, API = active pharmaceutical ingredient. *p*-values < 0.10 displayed, *p*-values < 0.05 were considered significant and are given in bold.

		Low Flow				High Flow			
API	LOQ	Pre-Ozonation	Difference	% Change	*p*-Value	Pre-Ozonation	Difference	% Change	*p*-Value
Alfuzosin	4	70.7	−20.5	−29.0	**0.0031**	86.3	−27.2	−31.5	0.0604
Amytriptyline	15	79.8	−6.0	−7.5		96.7	4.8	5.0	
Atenolol	15	1107.6	21.7	2.0		834.8	54.9	6.6	
Atorvastatin	15	675.6	−360.6	−53.4	**<0.0001**	898.8	−604.9	−67.3	**<0.0001**
Bisoprolol	4	518	−77.6	−15.0		369.4	5.1	1.4	
Bupropion	4	45.5	−2.2	−4.8		52.8	4.8	9.1	
Carbamazepin	7.5	175.7	−62.8	−35.7	**0.0156**	168.2	−13.1	−7.8	
Ceterizine	15	358.4	−90.1	−25.1	**0.0121**	368.5	−55.2	−15.0	
Ciprofloxacin	15	48,695.3	−12,400.2	−25.5		66,815.3	1526.1	2.3	
Citalopram	20	639	−282	−44.1	0.0816	550.3	−140.3	−25.5	
Clarithromycine	3	37.1	−20.2	−54.4		70.8	−0.6	−0.8	
Clindamycine	3	523.6	−119.3	−22.8	**0.0156**	476.6	−38.1	−8.0	
Codeine	20	783.6	−264.8	−33.8	**<0.0001**	901.6	−138.9	−15.4	
Diclofenac	15	411.2	−61.8	−15.0	0.0725	366.1	−22.5	−6.1	
Fexofenadine	10	104.9	−17	−16.2		101.5	9.6	9.5	
Flecainide	2	160.8	−10.4	−6.5		257.1	32.9	12.8	
Fluconazole	7.5	201.3	−2.7	−1.3		280.5	27.7	9.9	
Fluoxetine	7.5	33.5	−19	−56.7	**<0.0001**	48.5	−15.3	−31.5	**0.0015**
Irbesartan	3	97.3	−17	−17.5		58.6	−2.1	−3.6	
Metoprolol	15	1224.9	0.8	0.1		1117.4	46.2	4.1	
Mirtazapine	20	711.1	−229.5	−32.3	**0.0249**	521.7	−35.2	−6.7	
Oxazepam	10	191.4	−39.8	−20.8		209.4	3.9	1.9	
Paracetamol	20	2,633,386	−839,221.7	−31.9	**0.0499**	2,501,226	72,804.2	2.9	
Propranolol	30	198.4	−74.8	−37.7	**0.0014**	178.1	−13.7	−7.7	
Rosuvastatin	3	554.5	−133.5	−24.1		650.6	−35.2	−5.4	
Tetracycline	30	4511.2	−2693.1	−59.7	**<0.0001**	3352.2	−552.2	−16.5	
Tramadol	20	242.1	−16.5	−6.8		195.8	14.8	7.6	
Trimethoprim	4	617.6	−261	−42.2	**0.0006**	558.8	−27.3	−4.9	
Venlafaxine	20	1164.4	−137.6	−11.8		1073.1	65.8	6.1	
Mean all APIs				−25.2				−6.0	
Mean antibiotics				−40.9				−5.6	

## Data Availability

The data presented in this study are available in Appendix A.

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
