# Peer review of "On-Site Pilot Testing of Hospital Wastewater Ozonation to Reduce Pharmaceutical Residues and Antibiotic-Resistant Bacteria"

_antibiotics, 2021, doi:10.3390/antibiotics10060684_

Round 1

Reviewer 1 Report

In this paper the authors have demonstrated the use of ozonation in a pilot study to reduce active pharmaceutical ingredients in untreated hospital wastewater. The analysis was based in a previous published method Ref 18. The article is well-written, and the conclusions are supported by the results of the study. To me this paper deserves publication after revision.

Specific comments:

  1. Only antibiotics are included among the 29 APIs presented in table 1,  The authors should reconsider the title of the paper and replace “antibiotics” by active pharmaceutical ingredients.
  2. It will be good to present a list of the 92 APIs analyzed as a supplementary material.

Author Response

Dear Reviewer,

Thank you very much for spending time to evaluate and improve our work! Below please find a point-by-point response to the comments.

Only antibiotics are included among the 29 APIs presented in table 1,  The authors should reconsider the title of the paper and replace “antibiotics” by active pharmaceutical ingredients.

Thank you for this suggestion; we agree that an updated title more clearly will reflect the content of the paper. We suggest ‘On-site pilot testing of hospital wastewater ozonation to reduce pharmaceutical residues and antibiotic-resistant bacteria’.

It will be good to present a list of the 92 APIs analyzed as a supplementary material.

We agree. We have uploaded a list of the APIs analyzed and their limits of quantification as supplementary material, and referred to it in Results.

Reviewer 2 Report

Use ozone to treat raw wastewater is an interesting topic.

Disinfection is usually qualified by CT (concentration x contact time). Authors tested the disinfection performance using high flow and low flow. Obviously, under the same ozone dosage, the low flow will result in a high removal performance. But authors should try to calculate the CT to accurately show the disinfection performance. This will also allow the comparison between different studies.

Author mentioned that one of the key constraints was the sampling as it was not practical to do the continuous sampling, what are the reasons for that? what is the proposed mitigation strategy for future study.

The sampling was limited, making it difficult to make conclusions.

Authors have mentioned a few times about difficulties in implementing this system due to a few practical issues. Can authors provide detailed description on the issues?

In Table 1, authors only provided p-values for some results, what are the reasons?

Author Response

Dear Reviewer,

Thank you very much for spending time to evaluate and improve our work! Below please find a point-by-point response to the comments.

Disinfection is usually qualified by CT (concentration x contact time). Authors tested the disinfection performance using high flow and low flow. Obviously, under the same ozone dosage, the low flow will result in a high removal performance. But authors should try to calculate the CT to accurately show the disinfection performance. This will also allow the comparison between different studies.

We agree that including the ozone dose and calculating CT would have been preferable. However, unfortunately the ozone dose could not be obtained from the manufacturer of the ozonation system despite sustained efforts from our side. So unfortunately we cannot provide more information than what is in the manuscript at present. What we can know for sure is that the ozonation level was constant so that the CT must have been twice as high during the low flow. We acknowledge that this is a limitation of our study but still feel that publishing the results is warranted.

Author mentioned that one of the key constraints was the sampling as it was not practical to do the continuous sampling, what are the reasons for that? what is the proposed mitigation strategy for future study.

There were two reasons why continuous sampling was not feasible during the present study: i) the sampling equipment did not function properly because of the pressurized system; and ii) the practical challenges (as discussed below) did not allow for the ozonation system to keep running for longer periods of time. For future studies mitigation strategies are to: i) get sampling equipment that are adapted to pressurized systems; and ii) find an ozonation system better suited to untreated sewage and improve the installation so that continuous running of the system is possible.

The sampling was limited, making it difficult to make conclusions.

As discussed above, we acknowledge the grab sampling strategy as a limitation of our study. However, we do believe that the relatively high number of samples still allows us to draw some important conclusions despite the high variation, as demonstrated by several significant reductions in API levels.

Authors have mentioned a few times about difficulties in implementing this system due to a few practical issues. Can authors provide detailed description on the issues?

This is how we describe the issues in Discussion: ‘Although a 0.8 mm drum filter was used before the water entered the ozonation step, enough particles passed the filter to quickly clog parts of the system, necessitating frequent backwashes and cleanings. Pumps and piping had to be readjusted to accommodate for particles in the water and low flow rates. Other practical challenges faced were leakage of ozone from the collecting tank and unforeseen changes of pressure in the system. Although we had the pilot system in place for one year, the practical challenges only allowed us to run the system for approximately 2 months in the planned way, and then only for periods of time while being actively supervised and maintained by a technician.’ Unfortunately, we find it hard to elaborate more on this description. We have tried to enhance readability by referring to the above description the first time ‘practical problems’ are mention in the Discussion and rewriting the first part of the Discussion. We hope this will be satisfactory.

In Table 1, authors only provided p-values for some results, what are the reasons?

To avoid crowding of the table and facilitate for the reader, we have chosen only to report p-values <0.1 (as stated in the table caption). We have now clarified in the table caption that the values in bold (<0.05) are considered significant. We hope that the reviewer and the editor finds this an acceptable strategy. If you want us to display all p-values in the table, we can of course revise the manuscript accordingly.

Round 2

Reviewer 2 Report

The authors have addressed my comments.